# Genomic Signatures of Local Adaptation under High Gene Flow in Lumpfish—Implications for Broodstock Provenance Sourcing and Larval Production

**DOI:** 10.3390/genes14101870

**Published:** 2023-09-26

**Authors:** Simo Njabulo Maduna, Ólöf Dóra Bartels Jónsdóttir, Albert Kjartan Dagbjartarson Imsland, Davíð Gíslason, Patrick Reynolds, Lauri Kapari, Thor Arne Hangstad, Kristian Meier, Snorre B. Hagen

**Affiliations:** 1Department of Ecosystems in the Barents Region, Svanhovd Research Station, Norwegian Institute of Bioeconomy Research, 9925 Svanvik, Norway; snorre.hagen@nibio.no; 2Akvaplan-Niva Iceland Office, Akralind 6, 201 Kópavogur, Iceland; odj@akvaplan.niva.no (Ó.D.B.J.); aki@akvaplan.niva.no (A.K.D.I.); 3Department of Biological Sciences, High Technology Centre, University of Bergen, 5020 Bergen, Norway; 4Matís, Vínlandsleið 12, 113 Reykjavík, Iceland; davidg@matis.is; 5GIFAS AS, Gildeskål, 8140 Inndyr, Norway; pat.reynolds@gifas.no; 6Akvaplan-Niva, Framsenteret, 9296 Tromsø, Norway; lauri.kapari@akvaplan.niva.no; 7Lumarine AS, Stadionveien 21, 4632 Kristiansand, Norway; tah@lumarine.no; 8Lerøy Aurora, Stortorget 1D, 9008 Tromsø, Norway; kristian.meier@leroyaurora.no

**Keywords:** 3RAD, adaptive variation, aquaculture, cleaner fish, *C. lumpus*, genotype–environment association, seascape genomics, selective breeding

## Abstract

Aquaculture of the lumpfish (*Cyclopterus lumpus* L.) has become a large, lucrative industry owing to the escalating demand for “cleaner fish” to minimise sea lice infestations in Atlantic salmon mariculture farms. We used over 10K genome-wide single nucleotide polymorphisms (SNPs) to investigate the spatial patterns of genomic variation in the lumpfish along the coast of Norway and across the North Atlantic. Moreover, we applied three genome scans for outliers and two genotype–environment association tests to assess the signatures and patterns of local adaptation under extensive gene flow. With our ‘global’ sampling regime, we found two major genetic groups of lumpfish, i.e., the western and eastern Atlantic. Regionally in Norway, we found marginal evidence of population structure, where the population genomic analysis revealed a small portion of individuals with a different genetic ancestry. Nevertheless, we found strong support for local adaption under high gene flow in the Norwegian lumpfish and identified over 380 high-confidence environment-associated loci linked to gene sets with a key role in biological processes associated with environmental pressures and embryonic development. Our results bridge population genetic/genomics studies with seascape genomics studies and will facilitate genome-enabled monitoring of the genetic impacts of escapees and allow for genetic-informed broodstock selection and management in Norway.

## 1. Introduction

Land- and marine-based farming of aquatic animals (aquaculture) is a lucrative industry that produces high amounts of seafood estimated at a record high 87.5 million tonnes, valued at USD 264.8 billion in 2022, to meet the increasing consumer demand [1]. Aquaculture production alleviates commercial fishing pressure on natural populations, and it is an increasingly important component of global food security [1,2]. One of the most industrialised fish farming enterprises in the world is the Atlantic salmon (*Salmo salar* L.) aquaculture industry in Norway [3,4]. Seven other countries also produce farmed Atlantic salmon and include (in descending order of production after Norway) Chile, UK, North America, Faroe Islands, Ireland, New Zealand, and Australia [5,6]. Currently, one of the major constraints of further expansion of Atlantic salmon production in sea-based farming (mariculture) is disease outbreaks, specifically an ectoparasitic infestation of Atlantic salmon by copepod sea lice, which causes economic losses in Norway worth USD 436 million per year [4,7,8].

Sea lice cause direct damage to farmed stocks by feeding on their hosts’ skin, mucous, and blood [9], with severe infestations causing skin erosion, physical damage, osmoregulatory failure, increased disease incidence, stress, and immunosuppression [10,11,12]. Disease outbreaks are primarily caused by the salmon louse (*Lepeophtheirus salmonis* K.; specialist; specific for salmonids) and the sea louse (*Caligus elongates* N.; generalist; less host-specific) [7,9,13], collectively referred to as “sea lice” [8]. Consequently, one of the key objectives in the Atlantic salmon aquaculture industry is the designing and integrating of pest management plans to minimise sea lice infestations in sea cages [4,7,14]. Numerous treatment methods have been developed and used over the years to combat infestations of salmon lice [7,15]. Sea lice control methods initially involved the use of chemical treatments [16,17] and then started using “environmentally friendly” alternative or supplement biocontrol methods, involving the use of “cleaner fish”, because of the concerns surrounding the development of chemical-resistant strains of sea lice [8,18,19]. Consequently, the use of “cleaner fishes” that eat attached pre-adult and adult lice stages directly off the Atlantic salmon as an alternative to chemotherapeutics is not harmful to either the salmon, the environment, or the consumer, making this approach a leading contender [8,14,20].

Aquaculture of the lumpfish has become a large, lucrative industry owing to the escalating demand for cleaner fish to minimise sea lice infestations in Atlantic salmon mariculture farms [21,22,23,24]. Consequently, after Atlantic salmon and rainbow trout (*Oncorhynchus mykiss* W.), lumpfish is currently the third most profitable marine aquaculture species in Norway [25,26]. However, the stocking of cleaner fish in salmon sea cages has raised increasing concerns regarding the genetic integrity of natural populations following (accidental) escape events of cultured cleaners [27,28]. Accumulating genetics and genomics studies on natural populations of the lumpfish reveal varying patterns of population genetic structure across spatial gradients in the pan-Atlantic [28,29,30,31,32,33]. At large spatial scales, short tandem repeat (STR)-based studies revealed the existence of three major genetic groups of lumpfish: (1) western Atlantic, including samples from USA, Canada, and Greenland, (2) eastern Atlantic, including samples from Iceland and Norway, and (3) the Baltic [29,30,31], a pattern later confirmed with a panel of genome-wide SNPs [33]. At finer scales, a genetic break among northwestern and southwestern populations in Greenland was found [30] while no such break was apparent between northeastern and southeastern lumpfish populations in Norway [28,32]. Although, a divergent subpopulation in mid-Norway was found to be highly differentiated from other Norwegian sampling populations [31]. Moreover, a subset panel of 139 ‘diagnostic’ SNPs revealed that there were at least two genetic clusters in Norway, although not entirely concordant with geography [33]. Further, evidence for the adaptation to local oceanographic features across the pan-Atlantic was also evident in lumpfish [33]. Indeed, the abiotic environment acts as a selective force on natural populations, shaping and maintaining variation in heritable traits (morphology, physiology, and behaviour) that provide significantly greater fitness in their local environment (i.e., local adaptation) when compared to populations from other environments [34,35,36,37]. Understanding the extent and scale of local adaptation is critical in determining how quickly and to what extent specific populations will respond to habitat changes, climate change, fisheries—or farming-induced evolution and interactions with a hatchery—or captive-reared counterparts [36,38,39,40].

Gene flow plays a complex and multifaceted evolutionary role in local adaptation as it has the capacity to either enhance or disrupt local adaptation, specifically by spreading advantageous (“adaptive”) alleles or by homogenizing the gene pool [41,42,43,44,45]. Population genetics theory predicts that if a diversifying selection is not strong enough to prevent the loss in adaptive alleles, homogenizing gene flow between adaptively differentiated populations will “swamp out” whatever differences evolved in response to local environmental conditions (i.e., gene swamping) [46,47] and can, at worst, transfer maladaptive alleles into differentially adapted recipient populations (i.e., migration meltdown) [47,48,49]. Mounting empirical evidence challenges the classic concepts of the “swamping” and “meltdown” effects of gene flow by demonstrating that not only can gene flow promote local adaptation, but adaptive polymorphisms can also be conserved within populations subject to high gene flow [43,44,45,50,51,52,53]. These contrasting effects of gene flow are a key part of the eco-evolutionary and conservation puzzle [44], where in the context of rising anthropogenic pressures on the natural world, understanding the role of gene flow in local adaptation has become critical for biodiversity conservation [44,51,54].

In order to further investigate the genetic basis of local adaptation in lumpfish along the Norwegian coastline, this study builds upon our previous research in population genetics. Our earlier work focused on the lumpfish population genetics using non-functional short tandem repeats (STRs) as genetic markers [28], followed by a subsequent study utilizing functional gene-associated STR markers [32]. In this current study, we aim to expand our analysis to the genomic level by examining genetic variation potentially associated with the local adaptation in lumpfish. To achieve this, we employ the triple-enzyme restriction-site-associated DNA sequencing (3RADseq) technique, as described by Bayona-Vásquez et al. [55], which provides genome-wide molecular data in the form of single nucleotide polymorphisms (SNPs) and copy number variants (CNVs). Specifically, the present study aimed to investigate whether local adaptation occurs in the face of high gene flow in the species and to identify potential environmental selective pressures that drive adaptation at a finer scale. We also hypothesised that selection coupled with stochastic oceanic dispersal and hierarchical spatial structure results in chaotic genetic patchiness, where lumpfish exhibits genetic differentiation between subpopulations at large spatial scales, with limited fine-scale geographic patterns of population differentiation.

## 2. Materials and Methods

### 2.1. Study Species

The lumpfish is a benthopelagic, cold-adapted marine species found throughout the North Atlantic that is characterised by a globose body, skin that is typically covered with tubercles, the pelvic fins modified to form a sucking disc, and the usual presence of two short dorsal fins [56,57,58]. Lumpfish are often found in low densities and dispersed over a vast geographical area, and adults are largely solitary with limited social interaction compared to schooling fish [56]. Studies on movement ecology show that lumpfish are capable of facultative iteroparity and display philopatry to specific spawning sites, i.e., reproductive philopatry [59,60]. Spawning occurs annually between late spring and early summer (April to May) in shallow coastal waters, where males establish territory and nesting sites prior to the arrival of females, and subsequent to their arrival and effective egg laying in the nests, males fertilise and guard the eggs until they hatch [56]. During brooding, the male is the sole provider of the care that is necessary for offspring survival (i.e., paternal care), and after hatching, the larvae attach to substrates (e.g., seaweed and floating seaweed clumps) via the sucking disc [56,58]. Juveniles remain in shallow water areas for approximately 6–12 months before gradually making their way to the feeding grounds offshore. Juveniles eat copepods, gammarids, and polychaetes [61], while adults eat crustaceans, ctenophores, jellyfish, and polychaetes [62].

### 2.2. Sampling Regime

We concentrated our lumpfish sampling in the northeastern Atlantic coastal region, with a particular emphasis on the Norwegian coast. Here, the coast of Norway stretches for nearly 2000 km from south to north, where coastal areas are characterised by complex bathymetry and topography [63,64,65]. The shelf along the coast has an average depth of 300 m and many fjords stretch in from the coast, some with deep basins reaching 600–1300 m, and further offshore is the much deeper Norwegian Sea. The Norwegian Coastal Current (NCC) is the principal oceanic current along the Norwegian coast. It is a surface water mass that originates in the south (in the Skagerrak Strait) and combines with freshwater runoff from Norwegian fjords as it progresses northwards along the coast [64,65]. The North Atlantic Current (NAC) is another important oceanic current that flows beneath the NCC and periodically intrudes into the bank, bringing warm, saline, and nutrient-rich waters into some offshore locations, such as the Norwegian Froan archipelago. We collected a total of 107 tissue samples from 10 sites spanning coastal Norway (Figure 1), and sampling details are largely described in Jónsdóttir et al. [28] and Whittaker et al. [31]. In addition, we included two outgroup sample areas: Western Greenland (*n* = 15) and Canada (*n* = 15) in the northwestern Atlantic (Figure 1). Samples consisted of a small fin clip or muscle tissue from 5 to 15 lumpfish per site (Table 1).

### 2.3. Genetic Data

#### 2.3.1. DNA Extraction, 3RAD Library Preparation, and Sequencing

We primarily utilised the same samples from previous studies to avoid sampling and temporal variation issues between studies [28,31], which was appropriate for our research objectives. For tissue samples from Canada, Greenland, and Norway (samples not part of Jónsdóttir et al. [28] and Whittaker et al. [31]), we used the DNeasy Tissue Kit following the manufacturer’s instructions (Qiagen, Venlo, The Netherlands) to extract total genomic DNA from each sample. Then, we quantified all DNA stocks with the Quantus™ Fluorometer using the QuantiFluor^®^ ONE dsDNA System (Promega, Promega, WI, USA). We normalised each DNA sample to 10 ng/µL in nuclease and ion-free water and stored the working stocks at −21 °C prior to 3RAD library preparation.

We prepared RADseq libraries using the Adapterama III library preparation protocol of Bayona-Vásquez et al. [55] (their Supplemental File SI), which is a modified version of double-digest (dd) RAD [66] that uses three enzymes for digesting genomic DNA (3RAD). This procedure reduces the amount of DNA required since the third enzyme inhibits the formation of adapter dimers and DNA chimeras during the simultaneous digestion and ligation reactions [55,67,68]. We surveyed the literature and identified the frequently used enzymes in RADseq experiments of fish, and we chose a pair of enzymes that were compatible with one of the 3RADseq designs (Table 1 in [55]). We selected *MspI* (C|CGG) as the frequent cutter and *BamHI-HF* (G|GATCC) as the rare cutter, which fit the 3RAD Design 2 with *ClaI* (AT|CGAT) as the third enzyme for suppressing phosphorylated ends in *MspI* recognition sites. A detailed account of the laboratory procedure of the 3RAD library preparation can be found within the Appendix A.

We included 16 technical replicates to estimate genotyping error rates within and between lanes. Cleaned and indexed library pools per design were sent to the Norwegian Sequencing Centre (NSC) for quality control and subsequent final size selection using a one-sided bead clean-up (0.7:1 ratio) to capture 550 bp ± 10% fragments, and the final paired-end (PE) 150 bp sequencing on one lane of the Illumina HiSeq 4000 platform.

**Figure 1 genes-14-01870-f001:**
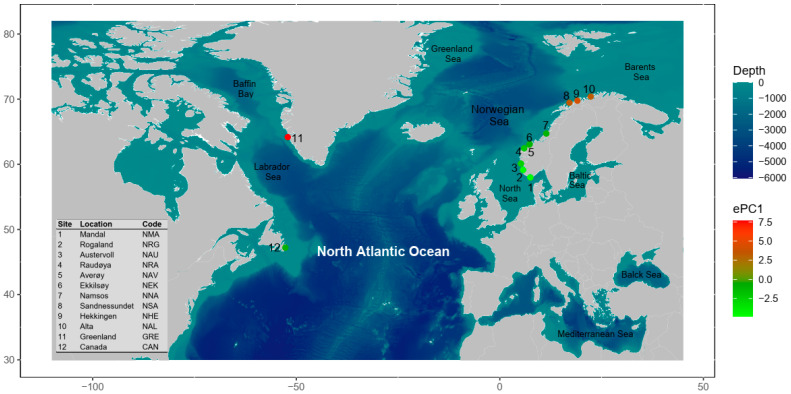
Sampling localities of lumpfish *C*. *lumpus* along the Norwegian coast in the northeastern Atlantic (Sites 1–10) and western Greenland (Site 11) and Canada (Site 12) in the northwestern Atlantic Ocean. Sampling sites are colour coded based principal component (PC) analysis (PCA) of environmental variables to describe patterns of the environmental variation in our sampling regime. The ePC1 represents the first PC loadings that explain the largest part of the total variance of 41%.

**Table 1 genes-14-01870-t001:** Information about sampling localities for lumpfish along the Norwegian coast and genetic variation descriptors: number of samples before filtering, followed by the number of individuals, which passed all filtering steps within parentheses (*N*), number of private alleles (*A_P_*), allelic richness (*A_R_*), observed heterozygosity (*H_O_*), gene diversity (*H_S_*), inbreeding coefficient (*F_IS_*) with 95% confidence interval, and sampling population-specific *F_ST_* (*β_WT_*) with 95% confidence interval.

Location	Code	Region	Latitude	Longitude	*N*	*A_P_*	*A_R_*	*H_O_*	*H_S_*	*F_IS_* (95% CI)	*β_WT_* (95% CI)
Mandal	NMA	Southern Norway	57.99	7.48	10 (9)	1	1.11	0.159	0.161	0.012 (0.005–0.020)	0.023 (0.014–0.031)
Rogaland	NRG	Southern Norway	59.15	5.70	11 (8)	1	1.15	0.155	0.151	−0.023 (−0.032–−0.015)	0.102 (0.092–0.113)
Hardangerfjord	NHA	Southern Norway	59.75	5.55	11 (0)	-	-	-	-	-	-
Austervoll	NAU	Southern Norway	60.10	5.19	7 (7)	0	1.17	0.158	0.165	0.041 (−0.011–0.013)	0.022 (0.010–0.033)
Raudøya	NRA	Southern Norway	62.45	5.97	5 (4)	0	1.17	0.163	0.165	0.014 (0.003–0.025)	0.020 (0.007–0.032)
Averøy	NAV	Southern Norway	63.01	7.23	10 (10)	4	1.16	0.162	0.163	0.006 (−0.001–0.014)	0.028 (0.020–0.037)
Ekkilsøy	NEK	Southern Norway	63.07	7.33	5 (4)	0	1.16	0.160	0.156	−0.022 (−0.034–−0.010)	0.066 (0.054–0.079)
Namsos	NNA	Central Norway	64.72	11.41	10 (10)	1	1.17	0.162	0.165	0.017 (0.010–0.024)	0.021 (0.013–0.030)
Sandnessundet	NSA	Northern Norway	69.76	19.05	12 (11)	3	1.16	0.159	0.156	−0.020 (−0.027–−0.013)	0.063 (0.055–0.071)
Hekkingen	NHE	Northern Norway	69.45	17.10	11 (11)	8	1.16	0.161	0.160	−0.008 (−0.015–−0.001)	0.047 (0.040–0.055)
Alta	NAL	Northern Norway	70.40	22.31	15 (14)	25	1.16	0.155	0.164	0.056 (0.050–0.063)	0.026 (0.018–0.033)
Sermersooq	GRE	Western Greenland	64.17	−52.09	15 (15)	243	1.15	0.151	0.154	0.020 (0.014–0.027)	0.087 (0.074–0.099)
Canada	CAN	Eastern (Atlantic) Canada	47.21	−52.69	15 (14)	180	1.13	0.135	0.135	−0.001 (−0.008–0.006)	0.188 (0.173–0.201)
Overall					137 (117)	38.8	1.15	0.157	0.159	0.008 (−0.004–0.013)	0.058 (0.054–0.061)

#### 2.3.2. SNP Discovery and Genotyping

After assessing the quality of the sequencing run post demultiplexing to plate-level using the plate-specific i7 indexes using FastQC [69] plus MultiQC v.2.31 [70], we processed the data as follows: First, we demultiplexed to the sample level using internal combinatorial barcodes and cleaned and trimmed the raw sequences with the perl script *process_radtags.pl* included as part of Stacks v.2.51 [71]. We ran the script using the “inline_inline” mode because the internal barcodes are part of the sequence in both Illumina reads. We removed reads with uncalled bases (-*c*) and discarded reads with low quality scores (-*q*) with a default sliding window of 15% of the length of the read and raw Phred score of 20 to retain high-quality reads. We specified Read 1 (*MspI*) and Read 2 (*BamHI-HF*) restriction enzymes, and we rescued sequence tags (internal tags) and RAD tags (enzyme overhang) within 2 mismatches of their expected sequence (-*r*); otherwise, reads were discarded. We truncated (-*t*) PE150 reads to 140 nt to have equal length among all reads with different barcodes. Second, we downloaded the reference genome of the lumpfish (accession number GCF_009769545.1; assembly version: fCycLum1.pri; scaffolds: 48; contigs: 395; N50: 4,950,682; L50: 35) from NCBI Genome database (https://www.ncbi.nlm.nih.gov/genome/; accessed on 10 March 2020) under BioProject PRJNA562003, the Vertebrate Genomes Project (G10K Consortium) [72]. Third, we then constructed the FM-index for the reference genome and individually aligned the cleaned, demultiplexed PE reads from each fish to the indexed reference genome using the BWA-MEM algorithm of BWA v.0.7.17 aligner [73], excluding reads with a minimum quality score of <30. Alignments were sorted and indexed, and read pairs were fixed using tools from the SAMtools v.1.9 suite [74]; we obtained alignment quality control (QC) statistics for the sorted and indexed alignments using BAMQC as implemented in Qualimap v.2.2.20 [75]. Finally, we used the alignments that passed QC to assemble RAD loci, call SNPs, and construct genotypes for individual fish using the perl script *ref_map.pl* also implemented in Stacks, which consisted of two components (*gstacks* and *populations*). We ran population analysis in *ref_map.pl*, with (i) a minimum of 75% of individuals in a population required to process a RAD locus for that population (-*r* 0.75), (ii) a minimum of 75% of populations a RAD locus must be present to process a locus (-*p* 9), (iii) by discarding unpaired reads (*--rm-unpaired-reads*), (iv) a minimum mapping quality of 20 to consider a read (*--min-mapq* 20), and (v) a minimum minor allele count (MAC) of 3 required to process a SNP at a RAD locus (*--min-mac* 3).

#### 2.3.3. Data Filtering

Recent research on marine species indicates that copy number variants (CNVs) are a significant source of genetic variability in genomes [76,77]. Therefore, we based our data filtering on the prior work of McKinney et al. [76] and Dorant et al. [77] to divide our 3RADseq dataset for the lumpfish containing all called SNPs from Stacks into two datasets, one containing single-copy loci (hereafter singleton SNPs) and the other containing multiple-copy loci (hereafter duplicated SNPs, CNV). We filtered genotype data and characterised singleton and duplicated SNP loci using filtering procedures and custom scripts available in STACKS Workflow v.2 (https://github.com/enormandeau/stacks_workflow; accessed on 25 March 2023). First, we filtered the ‘raw’ VCF file keeping only SNPs that (i) showed a minimum depth of four (-*m* 4) and (ii) were called in at least 80% of the samples in each site (-*p* 80), (iii) for which at least two samples had the rare allele, i.e., Minor Allele Sample (MAS; -*S* 2), using the python script 05*_filter_vcf_fast.py*. Second, we excluded those samples with more than 20% missing genotypes from the dataset. Third, we calculated pairwise relatedness between samples with the Yang et al. [78] algorithm and individual-level heterozygosity in vcftools v.0.1.17 [79]. Additionally, we calculated pairwise kinship coefficients among individuals using the KING-robust method [80] with the R package *SNPRelate* v.1.28.0 [81]. Then, we estimated genotyping error rates between technical replicates using the software tiger [82]. Finally, we removed one of the pairs of closely related individuals exhibiting the higher level of missing data along with samples that showed extremely low heterozygosity (<−0.2) from graphical observation of individual-level heterozygosity per sampling population. Fourth, we conducted a secondary dataset filtering step using 05*_filter_vcf_fast.py*, keeping the above-mentioned data filtering cut-off parameters (i.e., -*m* = 4; -*p* = 80; -*S* = 3). Fifth, we calculated a suit of four summary statistics to discriminate high-confidence SNPs (singleton SNPs) from SNPs exhibiting a duplication pattern (duplicated SNPs): (i) median of allele ratio in heterozygotes (*MedRatio*), (ii) proportion of heterozygotes (*PropHet*), (iii) proportion of rare homozygotes (*PropHomRare*), and (iv) inbreeding coefficient (*F_IS_*). We calculated each parameter from the filtered VCF file using the python script 08*_extract_snp_duplication_info.py*. The four parameters calculated for each locus were plotted against each other to visualise their distribution across all loci using the R script 09*_classify_snps.R*. Based on the methodology of McKinney et al. [76] and Dorant et al. [77], and by plotting different combinations of each parameter, we graphically fixed cut-offs for each parameter. Sixth, we then used the python script 10*_split_vcf_in_categories.py* for classify SNPs to generate two separate datasets: the “SNP dataset”, based on SNP singletons only, and the “CNV dataset”, based on duplicated SNPs only. Seventh, we postfiltered both datasets by keeping all unlinked SNPs within each 3RAD locus using the 11*_extract_unlinked_snps.py* script with a minimum difference of 0.5 (*-diff_threshold* 0.5) and a maximum distance 600 bp (*-max_distance* 500) to consider linked SNPs. Then, for the “SNP dataset”, we filtered out SNPs that were located in unplaced scaffolds, i.e., contigs that were not part of the 25 chromosomes of the lumpfish genome. We used PGDSpider v.2.1.1.5 [83] to transform some VCF files into input files for multiple programs. Lastly, to construct the CNV dataset, we extracted the locus read depth of SNPs identified as duplicated using vcftools and performed normalisation using the trimmed mean of *M*-values method originally described for RNAseq count normalisation and implemented in the R package *edgeR* [84]. The correction adjusts for the fact that an individual with a higher copy number at a particular locus will contribute proportionally more to the sequencing library than an individual with a lower copy number at that locus. The resulting CNV dataset comprised a matrix of normalised read counts for each individual at each CNV locus [77].

#### 2.3.4. Genome Scans and Signatures of Selection

We investigated if SNPs were putatively under selection using a population-level method (BayeScan) and two individual-level methods (*pcadapt* and *OutFLANK*). For BayeScan v.1 [85,86], we set the prior odds (*pr_odds*) at 10,000 after explorative runs (*pr_odds* = 100; 1000), which is appropriate for the number of markers in our datasets [86,87], and ran the model using 10,000 iterations with a thinning interval of 10, a burn-in of 200,000 steps, and 20 pilot runs of 5000 iterations. Then, we assessed convergence by visual inspection of trace plots, and additional diagnostic tests were carried out to confirm convergence, such as obtaining estimates of autocorrelation and effective sample size, Geweke’s convergence diagnostic, and Geweke–Brooks plots [88] with the R-package *coda* v.0.19.4 [89]. For further post-processing of BayeScan output in R, we defined a false discovery rate (FDR) *q*-value threshold of 0.05 and defined the respective *α* and *q* combinations to classify SNPs: (i) positive *α* and *q* ≤ 0.05 as suggestive of diversifying selection, (ii) negative *α* and *q* ≥ 0.05 as suggestive of balancing selection, and (iii) positive *α* and *q* ≥ 0.05 as suggestive of neutrality. For the respective individual-level methods, we used *pcadapt* v.4.3.3 [90] and *OutFLANK* v.0.2 [91] to carry out the analyses, and for each, we examined screen plots to determine the number of populations (*K)* and used the first 10 components that captured the majority of the population structure in the data. To control the false positive, we set the *F_ST_* threshold value to 0.05 in the *OutFLANK* approach and the FDR threshold value to 0.05 in the *pcadapt* approach. While BayeScan is suitable for our population-level sampling design, *PCAdapt* and *OutFLANK* are more reliable for species with complex, hierarchical population structure and are less sensitive to admixed individuals and outliers in the data [85,90,92]. Thus, to derive a putatively neutral SNP dataset for estimating population genetics parameters for GEA analysis, we removed outliers identified in any of the three methods to be putatively under selection.

#### 2.3.5. Genetic Diversity and Differentiation

To characterise genetic diversity, we calculated the average observed heterozygosity (*H_O_*), and within population gene diversity (*H_s_*), Wright’s inbreeding coefficient (*F_IS_*), and a population-specific index of differentiation relative to all other sampling populations (*β_WT_*) [93] using the R package *hierfstat* v.0.5-11 [94,95]. We computed the 95% confidence intervals of *F_IS_* and *β_WT_* values using the bootstrap method (1000 bootstraps) as implemented in *hierfstat*. We also calculated the number of private alleles at each locus and sampling population with the function ‘*privateAlleles*’ of the R package *StrataG* v.2.5.01 [96]. We calculated genome-wide pairwise *F_ST_* estimates using the Weir and Cockerham [97] method with the function ‘*pairwise.WCfst*’ in *hierfstat* and computed the 95% confidence intervals of *F_ST_* values using the bootstrap function ‘*boot.ppfst*’ (*nboot* = 1000) as implemented in *hierfstat*. Indices were considered significant if the 95% credible set did not include zero. We estimated the population genetic differentiation of loci identified as CNVs by calculating the variant fixation index *V_ST_* [98], an analog of the *F_ST_* estimator of population differentiation [97], commonly used to identify differentiated CNV profiles between populations and species [77,98,99,100]. For each pairwise population comparison, we calculated *V_ST_* as follows:$$V_{ST}=\frac{\left(V_{T}-\left(V_{i}\times N_{i}+V_{j}{\times N}_{j}\right)\right)/N_{T})}{V_{T}}=\frac{V_{T}-V_{S}}{V_{T}}$$
where *V_T_* is the total variance of normalised read depths among all individuals from populations *i* and *j*; *V_i_* and *V_j_* are the normalised read depth variance for populations *i* and *j*, respectively; *N_i_* and *N_j_* are the sample size for populations *i* and *j*, respectively; *N_T_* is the total sample size; and -*V_S_* is the average of the variance within each population, weighed for population size [77,98,100]. We performed permutation tests on the normalised read depths, where we randomly permuted all population *i* and *j* individuals and calculated a new *V_ST_* for every CNV. This process was repeated 1000 times creating a distribution of *V_ST_* values for each CNV using the R script available at https://github.com/DaRinker/PolarBearCNV (accessed on 10 February 2023). We calculated 95% confidence intervals, and indices were considered significant if the 95% credible set did not include zero. We also established genome-wide standard cut-offs of *V_ST_* > 0.1 < 0.3 and *V_ST_* > 0.3 based on the results from population comparisons of a marine species [77] and available species comparisons [100], respectively, to identify outlier CNV likely under selection.

#### 2.3.6. Genetic Clustering and Connectivity

To investigate population clustering, we first computed a matrix of Nei’s genetic distance *D_A_* [101,102] for each pair of populations with the function *stamppNeisD()* of the R package *StAMPP* v.1.6.3 [103]. The distance matrix was then used to perform a principal coordinate analysis (PCoA) using the function *pcoa()* of the R package *ape* v.5.6. [104]. Second, we examined genetic clustering with Discriminant Analysis of Principal Components (DAPC) using the R package *adegenet*v.2.1.5 [105,106], both with and without prior population assignment. We first used the adegenet function *find.clusters()* testing up to 30 clusters and using Bayesian information criteria (BIC) to identify the best-fit number of clusters for our data. Using the number of clusters with the lowest BIC value, we performed a DAPC with the adegenet function *dapc()* and plotted the results in R. The number of PCs was set based on the *a*-score. Third, we inferred the genomic admixture of lumpfish using a sparse non-negative matrix factorisation (snmf) as implemented in the *snmf()* function in the R-package LEA v. 3.8.0 [107,108]. We calculated estimates of individual admixture coefficients over a range of *K* values (1–10). We determined the number of ancestral populations (*K*) using the entropy criterion [107,109]. Ancestry coefficients (*Q*) were visualised by plotting the *Q*-values of each individual in a bar plot using the R package *pophelper* v.2.3.1 [110]. Last, we assessed the genomic co-ancestry among individuals with fineRADstructure [111]. We first inferred a co-ancestry matrix with the script *RADpainter*. Subsequently, clustering was performed with the Markov chain Monte Carlo method of fineRADstructure, running for 500,000 generations and sampling every 1000 generations; the first 200,000 generations were discarded as burn-in (non-default parameters: -*x* 200,000 -*y* 300,000 -*z* 1000). We also inferred a tree for visualisation with fineRADstructure using the tree-building algorithm of Lawson et al. [112] with 10,000 attempts (non-default parameters: -*m* T -*x* 10,000). fineRADstructure results were plotted with R scripts included in the fineRADstructure package. We only used PCAs to visualise the pattern of genetic differentiation based on all CNVs in R.

### 2.4. Environmental Data

Seascape genomics analyses require a comprehensive characterisation of the marine environment in order to avoid the confounding influence of collinear gradients [113,114,115]. As a result, the seascape characterisation we employed included 16 environmental variables, which include climate and biogeochemical variables: sea surface temperature (SST), sea surface salinity (SSS), eastern sea current velocity (ESCV), northern sea current velocity (NSCV), chlorophyll concentration (CHLa), chlorophyll mass concentration (CHLm), phytoplankton concentration (PHYC), net primary production (PP), suspended matter (SPM), dissolved oxygen (O_2_) concentration (hereafter abbreviated as DO), nitrate (NO_3_) concentration (hereafter abbreviated as NO3), phosphate (PO_4_) concentration (hereafter abbreviated as PO4), silicate (SiO_4_^4−^) concentration (SI), dissolved iron (Fe) concentration (hereafter abbreviated as FE), surface partial pressure of carbon dioxide in sea water (*p*CO_2_^sea^; hereafter abbreviated as SPCO2), and pH (hereafter abbreviated as PH). We obtained these variables from five respective georeferenced datasets describing atmospheric and seawater conditions from the Copernicus Marine Environment Monitoring Service (CMEMS: https://marine.copernicus.eu/, accessed on 10 November 2022; Appendix A). We obtained the average monthly values down to 156 m in depth between 1993 and 2014, with a spatial resolution ranging from ~9 to 4 km, from CMEMS, before the genetic data were sampled (2015–2017) [28,31]. We chose to acquire average daily values for SST, SSS, ESCV, NSCV, and CHLa because these variables are typically the main environmental drivers of local adaptation in the marine environment for species with a planktonic larval phase [116,117,118,119].

We processed these variables in the R environment using the *raster* package v.2.8 [120] to compute the overall mean (OM) and standard deviation (OSD) associated with the mean for each variable. CMEMS provides information on sea current velocity in two separate datasets, ESCV and NSCV, which represent the water movement towards east and towards north, respectively. Therefore, prior to computing the overall mean and standard deviation of the overall sea current velocity (SCV), we considered the ESCV and NSCV as perpendicular vectors of water velocity, where the Euclidean norm of these vectors is the overall SCV. We computed the overall SCV as the square root of the sum of squares of the ESCV and NSCV.

Moreover, given the importance of temperature in local adaptation, we also computed four seasonal statistics for SST, which include the mean and standard deviation temperature of the coldest and hottest month, respectively. In total, we computed 34 environmental variables, which we used to summarise the patterns of environmental variation across sampling sites using principal component and hierarchical clustering analyses in base R with the *prcomp()* and *hclust()* functions.

In order to avoid multicollinearity and select the most seemingly ecologically relevant variable, we used the *corSelect()* function of the R program *fuzzySim* v.4.3 [121] with a threshold value of the correlation coefficient −0.5 ≤ |*r*| ≥ 0.5 and excluded the variable with the highest Variance Inflation Factor (VIF) [122,123] among correlated variables. Nevertheless, we identified groups of highly collinear environmental variables using a hierarchical clustering implemented in the R package *KlaR* v.1.7-2 [124] for the downstream interpretation of genotype–environment associations to facilitated comparison with other studies.

### 2.5. Spatial Data

We implemented the spatial eigenfunction analysis (SEA) and isolation-by-distance (IBD) approaches to evaluate the relative contribution of the geographic distance on multiscale spatial patterns of genetic variation in our system. For both analyses, we used the *F_ST_*-based genetic distance matrix calculated from the neutral SNP datasets consisting of the (i) global (Norway, including outgroups) and (ii) local (Norway) sampling sites. For SEA, we used distance-based Moran Eigenvector Maps (dbMEM) [125,126] to model spatial structure across our sampling sites. Briefly, dbMEMs represent structure in a dataset with orthogonal axes to describe complex patterns of spatial structure in rectangular form suitable for use as a cofactor to control for spatial correlation in downstream statistical tests of y~x relationships, e.g., partial redundancy analysis (RDA) in GEA studies [127,128]. Importantly, these independent vectors can model both broad- and fine-scale patterns, which may affect genetic variation partitioning across large and/or highly heterogeneous landscapes. To calculate dbMEMs, we first converted degrees latitude and longitude to Cartesian coordinates with the *geoXY()* function available in the R package *SoDA* v.1.0-6.1 [129]. Then, we computed a Euclidian distance matrix on the Cartesian coordinates using the *dist()* function and also transformed geographical distances as in-water distances using the *lc.dist()* function of the R package *marmap* v.1.0.6 [130]. We generated two sets of dbMEM variables by decomposing (i) Euclidean (geographic) and (ii) in-water (least-cost) distances between sites, respectively, with the *dbmem()* function implemented in the R package *adespatial* v. 0.3-20 [131]. We used the *cor.test()* function to determine the correlation between ‘Euclidean’ and ‘in-water’ dbMEMs. Last, we performed two site-level RDAs on sites’ allelic frequencies and geographical distances using (i) in-water distances and (ii) significant dbMEM with the *rda()* function. To test for IBD, we used the *mantel.rtest()* function of the R package *ade4* v.1.7-20 [132] with 10,000 permutations using Euclidean and in-water distances.

### 2.6. Seascape Genomics

We carried out a seascape genomics analysis to investigate the possible correlation between environmental variables and the frequency of particular genotypes of lumpfish globally and regionally. Associations of this kind might reveal an environmental constraint requiring adaptation in lumpfish at different scales, as well as the genetic features conferring the selective advantage. For genotype–environment association analyses, we combined univariate and multivariate methods, namely latent factor mixed models (LFMM) and redundancy analysis (RDA). While LFMM identifies associations between single loci and single predictors, RDA can detect multilocus signatures of selection as a function of a multivariate set of predictors [133,134]. The LFMM and RDA approaches have been used extensively in the field [135,136] and provide a good compromise between detection power and error rates and are robust to a variety of sampling designs and underlying demographic models [113,133]. Both methods assume a linear relationship between allele frequency and environmental variables and require complete datasets (without missing values). For this reason, we imputed missing genotypes, however, using the most common genotype at each SNP across all individuals instead of ancestry-based missing data imputation given that our study species is characterised by genetic homogeneity between many sampling populations and the low levels of missing data (<5% in our datasets, see Results).

To apply LFMM, we first used the R function *rda()* function in vegan v. 2.6-4 [137] to perform a principal component analysis (PCA) on environmental predictor variables to generate synthetic predictors, instead of using raw predictor variables, in order to minimise the number of tests. We determined the optimal number of principal components (PCs) to retain using the broken-stick criterion [138] with the *screeplot()* function in vegan, which suggests retaining only those components whose eigenvalue is larger than the value given by the broken-stick distribution [138,139]. We used the *scores()* function as implemented in vegan to determine correlations between the PC axis and predictors. Similarly, we applied the same approach to estimate population structure in the genotypic data and determined the optimal number of principal components (PCs) to retain. After determining how many PCs explained most of the genotypic variance (*K*, for setting latent factors) and predictor variance, we conducted the LFMM analysis with the newer *lfmm2* implementation that uses a least-squares estimation method to compute LFMMs using the R package *lffm* v.1.1 [134]. We implemented the *lfmm_ridge()* function for estimating the parameters of LFMMs, where the ridge estimates are based on minimising a regularised least-squares problem with an *L*_2_ penalty [134]. We then post-processed the model outputs for each PC with the *lfmm_test()* to perform association testing using the fitted model and calibrated *p*-values using the *gif* option that calculated the genomic inflation factor (GIF; λ). The *lfmm_test()* function returns *p*-values describing the statistical significance of every genotype–environment association under different values of *K*. For each association model related to the same environmental variable, we converted GIF-adjusted *p*-values to *q*-values and applied the FDR control method to correct for multiple testing with the *qvalue()* function of the R package *qvalue* v.2.28.0 [140], and the values were deemed significant if *q* < 0.05 under at least one level of *K*.

We then performed individual-level RDA using the *rda()* function for modelling genotypes as a function of predictor variables and produced as many constrained axes as predictors [133]. We assessed the significance of RDA constrained axes using the ‘*ANOVA.cca*’ function, and we then used the significant axes to identify candidate loci. Candidate loci were identified using a Mahalanobis distance-based approach [141], which made the RDA result comparable with those obtained with LFMM since it allowed adjusting *p*-values using the genomic inflation factor (*λ*) and setting FDRs to *q* = 0.05, as described above.

### 2.7. Functional Annotation Analysis

Given that landscape/seascape genomics analyses can suffer from the issue of false positives, we annotated the genomic neighbourhood of each SNP associated with a particular environmental variable(s), i.e., environment-associated SNP, and determined whether the molecular functions of the genes surrounding an environment-associated locus were consistent with an assumed adaptive role. For every environment-associated SNP, we performed the annotation procedure as follows. First, we annotated environment-associated SNP with the NCBI General Feature Format (GFF) annotation file of the reference genome of the lumpfish using SnpEff v.5.1 [142], and we retrieved all the predicted genes falling within the ±50 kilobases (kb) window. This maximum window size was chosen because genes associated with a mutation can be found hundreds of kbs away [143,144,145]. Second, we retrieved the predicted protein sequences related to these genes and ran a similarity search with BLASTp [146] against metazoan protein sequences in the UniProtKB/Swiss-Prot database (Release 2023_01) [147]. For every predicted gene, we kept only the best significant match with an E-score threshold < 10^–7^. Lastly, we performed pathway and process enrichment analysis for Gene Ontology terms, KEGG Pathway, Reactome Gene Sets, and WikiPathways among the predicted genes using the Metascape web server v.3.5.20230101 [148].

## 3. Results

### 3.1. Genomic Datasets and Signatures of Selection

Although our HiSeq4000 run yielded approximately 330 million PE reads, the total number of raw reads reported in this study is 162,423,062, and the rest of the reads correspond to those from other studies on the species in our group. Processing of raw Illumina data by the program *process_radtags* recovered a high percentage of retained reads (92.7%) with a mean of 1.2 million reads per sample. After removing samples that did not sequence well, with <500,000 retained reads, the *gstacks* program generated a catalog of 457,751 putative RAD loci with a mean insert length of 442.8 bp (±155.7 bp) and a mean effective per-sample coverage of 18× (±14.3×). After the initial SNP filtering with the *populations* program, we retained 57,840 RAD loci composed of 40,972 SNPs for the global dataset and 60,540 RAD loci composed of 40,877 SNPs for the local dataset. After secondary filtering, thinning the step in STACKS Workflow, purging closely related individuals, and removing SNPs within non-chromosomal scaffolds, we retained a final panel of 21,202 and 14,613 polymorphic SNPs for the global (genotyping rate = 0.95; *n* = 117) and regional (genotyping rate = 0.98; *n* = 88) datasets, respectively (Appendix A). We detect 5053 putative SNPs under balancing selection and 9 SNPs putatively under divergent selection with BayeScan, 213 putative SNPs under divergent selection with *pcadapt* and 0 SNP putatively under divergent selection with *OutFLANK*. We derived a putatively adaptive SNP dataset comprised of outlier loci under divergent selection. We removed SNPs putatively under selection and retained datasets of 15,953 and 10,153 putatively neutral SNPs for the global (genotyping rate = 0.95; *n* = 117) and regional (genotyping rate = 0.98; *n* = 88) sampling populations. From these datasets each, we derived a SNP subset dataset of 150 SNPs with the highest locus-specific *F_ST_*. We identified 803 CNV, and after normalizing the data and removing all CNV with a variance of zero, we retained a final panel of 27 CNV for each dataset.

### 3.2. Genetic Diversity and Differentiation

The overall patterns of the genome-wide diversity dataset for global and regional samples were similar across sampling populations, with *H_O_* and *H_S_* ranging from 0.135 to 0.163 and 0.135 to 0.165, and *F_IS_* ranging from −0.023 to 0.056 (Table 1). Most notably, *A_P_* patterns among global and regional sampling population varied, ranging from 0–243, where lumpfish from the northwestern Atlantic (CRE and CAN) harboured the highest *A_P_* of 243 and 180 alleles. Within Norway, the northernmost sampling populations NAL and NHE harboured the highest *A_P_* of 25 and 8. The overall *β*_WT_ index for the global dataset was moderate (0.058; 95% CI: 0.054–0.061), where the highest observed *β_WT_* within sampling populations was among lumpfish from NRG (0.10, 95% CI: 0.092–0.113) and CAN (0.188, 95% CI: 0.173–0.201). Whereas the overall *β*_WT_ index for the regional dataset was shallow (0.012; 95% CI: 0.009–0.015), where the highest observed *β_WT_* was among lumpfish from NRG (0.065, 95% CI: 0.053–0.076) and NSA (0.015, 95% CI: 0.007–0.024).

We uncovered shallow-to-great population differentiation among sampling populations with the global dataset, with pairwise *F_ST_* ranging from −0.002 to 0.182 (global *F_ST_* = 0.072), where outgroup sampling populations were the most divergent followed by the southern Norway sampling population NRG (Table 2). We detected shallow population differentiation among Norwegian sampling populations with the regional dataset similar to the global dataset, with pairwise *F_ST_* ranging from −0.001 to 0.036 (global *F_ST_* = 0.011), where NRG and NAU were the highly differentiated sampling populations. We found a similar trend of a limited discernible geographic pattern with the CNV dataset (Table 2). We uncovered similar patterns of genetic diversity and population differentiation with the subset dataset of 150 SNPs with the highest locus-specific *F_ST_* and ‘adaptive’ SNP dataset (Appendix A).

### 3.3. Population Genetic Structure and Connectivity

We explored population structure using several independent methods, and we did detect genetic structuring with the global, regional, and subset SNP datasets. First, globally, the PCoA revealed geographic clustering of lumpfish from the three different regions (Canada, Greenland, and Norway), where samples from respective regions formed individual clusters with a hierarchical pattern of genetic structuring within Norway. Overall PCoA clustering patterns revealed at least three main genetic clusters based on the first three principal components (PCs), which together accounted for 26% of the genetic variation in the global datasets (Figure 2A). A similar clustering pattern was recovered with the regional 150-SNP dataset (Figure 2B). Whereas at least two main genetic clusters were evident in Norway with the regional dataset, the first three PCs together accounted for 9% of the genetic variation (Figure 3A). Although, the regional 150-SNP dataset recovered finer scale patterns of genetic clustering with the first three PCs accounting for 52.9% of the genetic variation (Figure 3B). Second, the DAPC analysis also uncovered similar geographical clustering patterns of lumpfish as a priori, and when excluding location prior to where three and two genetic clusters were identified based on the BIC score. The genetic clusters corresponded with geography at the global scale but not at the regional scale, where Norway constituted two sub-clusters. Third, the post-processing of the *snmf* results based on the lowest average CV score across replicates identified *K* = 3 and *K* = 1 as the most likely number of genetic clusters for the global and regional levels, respectively, which was consistent with the number of clusters based on the first two principal components and DAPC, respectively (Figure 4A,B). Notably, the genetic composition of NRG was largely consistent among the genetic differentiation estimates, where it was the most divergent sampling population of lumpfish in Norway (Figure 4A,B). Finally, the fineRADstructure co-ancestry between genotype pairs revealed geographic clustering of genotypes by region (Canada, Greenland, and Norway) and the northwestern Atlantic sampling populations shared more co-ancestry with each other than between Norwegian sampling populations in the northeastern Atlantic (Figure 5). Genotypes in Norway were weakly clustered, and the separation into two subgroups was marginally supported, where southern Norway individuals from NMA, NRA, and NAU largely constituted one group (Figure 5). Moreover, we uncovered that there was a finer scale of genetic structuring observed from lumpfish collected from the northernmost Altafjord.

In contrast to the SNP datasets, the global and regional CNV data revealed three and two clusters, respectively, which do not correspond to the geographic proximity among the sampling population (Figure 6 and Figure 7). The global dataset revealed that GRE, NNA, and NRA were differentiated from the rest of the study sampling populations (Figure 6A–C). The regional dataset corroborated the global dataset with regard to the differentiation of NNA and NRG (Figure 7A–C).

### 3.4. Spatial Structure and Environmental Associations

We found a strong, significant correlation between pairwise *F_ST_* and Euclidean geographic distance (Mantel *r* = 0.947, *p* = 0.0041) and in-water geographic distance (Mantel *r* = 0.952, *p* = 0.0029) with the Mantel test, indicating a significant effect of IBD in the global dataset. However, within Norway, we detected no significant effect of IBD with the regional dataset (Euclidean: Mantel *r* = 0.110, *p* = 0.2767; in-water: Mantel *r* = 0.166, *p* = 0.217). We retained the first four axes from the PCA of Hellinger-transformed allele frequencies, which cumulatively explained 56.8% of the total neutral genetic variation for the Norwegian sampling populations. Forward selection of the Euclidean dbMEM variables identified two significant predictors (dbMEM6 and dbMEM8) representing regional-scale spatial structure. RDAs with only the selected dbMEM variables were significant (*p* = 0.013) with a coefficient of determination (adjusted *R*^2^) of 22.3%. The first two RDA axes were marginally significant and explained 20.8% and 18.8% of the genetic variation summarised in the PC axes, respectively (Figure 8).

For seascape genomics analysis, we found that our sampling regime constituted of at least four environmental clusters based on principal component and hierarchical clustering analyses (Figure 1). Our sampling scheme harboured adequate environmental gradients for both global and regional seascape genomic analysis. We identified seven independent environmental variables based on an analysis of the correlation among environmental variables for the global sampling populations, whereas four independent variables were identified for the regional sampling populations. For the global dataset, we detected 542 significant genotype–environment associations (with 5 duplicate detections; 537 unique GEAs) based on RDA analysis without accounting for spatial structure using partially constrained ordination (Figure 9A,B). Environment-associated SNPs were related to CHLm_OM (151 SNPs), CHLm_OSD (53 SNPs), FE_OSD (48 SNPs), CHLa_OM (82 SNPs), MEM4 (107 SNPs), O2_OSD (45 SNPs), and SCV_OM (51 SNPs). While 382 unique environment-associated SNPs were detected when accounting for spatial structure with pRDA (Figure 10A,B): CHLm_OM (131 SNPs), CHLm_OSD (69 SNPs), FE_OSD (15 SNPs), CHLa_OM (64 SNPs), MEM4 (12 SNPs), O2_OSD (50 SNPs), and SCV_OM (41 SNPs). Using LFMM, we identified 33 environment-associated SNPs with the global dataset. Regionally along the Norwegian coastline, we detected 70 significant genotype–environment associations with RDA (CHLm_OSD: 1 SNP; FE_OM: 42 SNPs; MEM6: 22 SNPs; MEM8: 5 SNPs), 48 with pRDA (CHLm_OSD: 13 SNPs; FE_OM: 33 SNPs; MEM6: 2 SNPs; MEM8: 0 SNPs), and 5 with LFMM. We did not detect any environment-associated CNVs with either the global or the regional dataset, but uncovered a similar pattern of genetic structure revealed by PCA analysis (Figure 11A,B).

### 3.5. Functional Annotation

Across GEA methodologies, we detected 654 and 106 unique environment-associated RAD loci with the global and regional datasets, respectively, with an overall of 35 loci (Appendix A). The functional annotations surrounding the global panel of environment-associated SNPs revealed that the top-level Gene Ontology (GO) categories were developmental process, metabolic process, localisation, cellular process, homeostatic process, response to stimulus, biological regulation, positive regulation of biological process, multicellular organismal process, and negative regulation of biological process (Appendix A). The genes were involved in four Reactome Gene Sets (phase 0—rapid depolarisation, EPHA-mediated growth cone collapse, gap junction trafficking and regulation, and SUMO transferred from E1 to E2 (UBE2I, UBC9)), whereas the regional panel of environment-associated SNPs were involved in the cellular process, developmental process, positive regulation of the biological process, response to stimulus, growth, biological regulation, metabolic process, and locomotion (Appendix A). The genes were involved in two KEGG pathways (MAPK signalling pathway and RIG-I-like receptor signalling pathway) and four Reactome Gene Sets (nucleotide-like (purinergic) receptors, lysosome vesicle biogenesis, postmitotic nuclear pore complex reformation, and signalling by PDGF)). The 35 overlapping loci were involved in three GO biological processes (cellular process, developmental process, and response to stimulus) and three KEGG pathways (Glycerophospholipid metabolism, WNT signalling, and Hedgehog signalling pathways) (Appendix A).

## 4. Discussion

It is widely accepted that local adaptation to the abiotic environment plays a major role in both intraspecific and interspecific diversification; however, its contribution relative to other evolutionary forces, such as gene flow is rarely quantified [44,52,54]. In this study, we investigated whether local adaptation occurs despite high gene flow in lumpfish, and if it does, we identified the principal environmental drivers of local adaptation at both larger and finer scales. We found associations between genetic and climatic structure in this system and quantified the contributions of individual environmental variables to patterns of genomic variation and dissected the genetic targets of these putative selective gradients in a multivariate statistical framework at a finer scale. Furthermore, population genomic analyses using genome-wide SNPs revealed that genetic homogeneity was prevalent in Norwegian sampling populations, correlating with previous findings using microsatellite markers [28,32]. This study is one of the largest population and seascape genomics efforts performed to date on the Norwegian lumpfish and is predicted to facilitates genome-enabled monitoring of the genetic impacts of escapees and allow for genetic-informed broodstock selection and management.

### 4.1. Genome Scans for Outliers

Genome scans for the selection indicated both signatures of divergent and balancing selection acting on the lumpfish genome. The putative genomic regions under divergent selection that we uncovered revealed that sampled populations from Norway were not too adaptively divergent. We found that the balancing selection was the predominant force operating on the lumpfish genome, indicating that multiple alleles were actively maintained at frequencies higher than expected from genetic drift alone [42,149]. The balancing selection, such as the negative frequency-dependent selection, plays a crucial role in maintaining allelic and phenotypic polymorphisms within populations, including behavioural traits, such as foraging [150], personality [151,152], behaviourally mediated life history strategies [153], and migration [154,155]. Our results represent the first genetic evidence for the balancing selection in lumpfish that could be linked to various commercially important traits, including sea lice foraging habits. Although, there have been issues surrounding the sustainability and delousing efficacy of wild-caught cleaner fish [20,156], mounting evidence suggests that delousing-related traits, such as small body size (40–140 g), personality, and grazing efficacy, are likely parentally controlled and can be optimised under selective and targeted breeding programs [26,157,158,159]. Accordingly, our results are foundational towards understanding whether selection operates differentially within populations, between individuals, or not for traits of interest, such as sea lice foraging behaviour.

### 4.2. Understanding the Population Structure of Lumpfish

Delineating the patterns and levels of population structure at various scales in lumpfish has major implications for conservation, fisheries, and aquaculture. With our ‘global’ sampling regime, we found two major genetic groups of lumpfish, i.e., the western (Canada and Greenland) and eastern Atlantic (Norway), congruent with previous findings [29,30,31,32,33]. Our results revealed that lumpfish from the western Atlantic (outgroup) group shared higher co-ancestry with each other than between populations from the eastern Atlantic group. Within the eastern Atlantic group, conflicting findings have been presented based on genomic STRs (g-STRs) [28,31], expressed sequence tag (EST) STRs (EST-STRs) [32], and genome-wide SNPs [33] on whether there is genetic structuring among Norwegian lumpfish. Our results revealed that Norwegian lumpfish were weakly clustered into two marginally supported subgroups, where individuals from southern Norway largely constituted one group. With both SNP and CNV, we found no evidence of population structure with sampling populations from Jónsdóttir et al. [28,32], suggesting that the lack of genetic structure in the previous studies was not preordained by the marker of choice, i.e., g-STRs [28] or EST-STRs [32]. In fact, a similar trend of genetic homogeneity was observed in Jansson et al. [33] between representative sampling populations from Norway (Flatanger and Flekkefjord) with genome-wide and ‘diagnostic’ SNPs datasets, where weak population differentiation and structure between populations was uncovered regardless of the SNP panel used and the large geographical distance between the sampling locations. Overall, our results of population genetic structure of the Norwegian lumpfish agree with the findings of the existence of one [28,32] and two [31,33] genetic groups depending on the finer scale coverage of the Norwegian coastline. Nonetheless, it is clear from these studies that population genetic structure in lumpfish is influenced by the interactions between evolutionary forces (gene flow and genetic drift), reproductive behavioural traits (philopatry), and seascape features (hydrodynamic, thermal, or eco-physiological boundaries).

### 4.3. Local Adaptation under High Gene Flow in Lumpfish

The intensity of divergent natural (or Darwinian) selection frequently fluctuates in heterogeneous environments across ecological clines (i.e., selective gradients), resulting in genotype–environment interactions for Darwinian fitness [160,161]. Marine species are largely characterised by the lack of apparent physical barriers to gene flow across vast geographic distances [118,162,163]. However, dispersal potential may differ over a fragmented seascape depending on patterns and gradients of environmental variables, such as ocean currents, temperature, pH, and salinity, resulting in varying levels of population connectivity [116,118,163]. Consequently, marine species provide an ideal system for understanding how diversifying selection operates on maintaining divergence between populations at adaptive loci, whilst allowing homogenisation in other parts of the genome, i.e., local adaptation with gene flow. Our results provided novel insights into the existence of local adaptation under high gene flow in the Norwegian lumpfish governed by selective gradients along the coastline, with temperature and salinity correlated variables and sea current velocity as the main environmental drivers of local adaption in lumpfish. Within Norway, we found that the northern and southern populations are differentially adapted to local conditions. These findings add to the growing body of knowledge that oceanographic currents aiding larval transport are among the major drivers of both genetic homogeneity and population structure in species with a passive dispersal phase depending on the oceanographic regime and spatial scale [28,33,117,118,119,164]. Moreover, our results were consistent with those of previous research demonstrating that gene flow can promote local adaptation and that adaptive polymorphisms can be conserved within populations subject to high gene flow, contrary to the traditional notion of the ‘swamping’ and ‘meltdown’ effects of gene flow [43,44,45,50,51,52,53]. Given the predominant genome-wide homogeneity in Norwegian lumpfish, our results uncovered that some parts of the lumpfish genome are targets of divergent selection while other parts are under the influence of high gene flow.

### 4.4. The Genomic Profile of Environmentally-Associated RAD Loci

Functional annotation analyses of the environment-associated RAD loci revealed that the gene sets linked to these loci play a key role in biological processes associated with environmental pressures. The retrieved top GO categories (e.g., cellular process, developmental process, metabolism, biological regulation, and response to stimulus) have been reported in other studies that measure transcriptomic responses under experimentally induced heat stress [165,166], ocean acidification [167,168], biomineralisation, energy metabolism, and heat, disease, or hypoxia tolerance [40,169]. Moreover, the gene sets reported in the present study were enrichment in four functional pathways, which included the WNT-, MAPK-, RIG-I-like receptor and Hedgehog signalling pathways and Glycerophospholipid metabolism. The WNT and Hedgehog signalling pathways play a crucial role in embryonic development [170,171,172], suggesting that divergent natural selection largely operates during larval development. In addition, the MAPK pathway is essential for the response to signals or stresses from changing environmental conditions and various other stimuli [173]. Whereas the RIG-I-like receptor pathway is involved in the regulation of the innate immune response against viruses [174]. Overall, these findings broaden our understanding of the genetic basis of local adaptation in natural populations and add to the growing database of functional annotated environment-associated adaptive loci.

### 4.5. Conclusions, Conservation, and Aquaculture Implications

In the upheaval of environmental change and commercial exploitation of the lumpfish, applied genomics is paramount for efficient conservation and sustainable species management. This study conducted analyses of population structure on all SNPs but also independently on putatively neutral and outlier SNPs to aid in the inference of neutral versus adaptive processes underlying the observed genetic patterns. Our study provides important clues on the definition and delineation of stocks in Norway and their potential to respond to environmental variability. Additionally, our analyses of population structure support the subdivision of lumpfish into at least two stocks in Norway, where one of the southernmost sampling populations, NRG, was the most divergent sampling population. Our regional sampling regime allowed us to resolve the contradictory evidence relating to genetic homogeneity versus structuring among Norwegian lumpfish and identify the likely causes for the conflicting results of the spatial scale of sampling design given the complex bathymetry, topography, and oceanography of the coastline. Moreover, the patterns of local adaptation in lumpfish uncovered in our study provide further valuable information that can help identify the best potential source populations for broodstock development and provenance sourcing and to identify escapees from farms.

Future sampling design should focus on embayment and/or fjord locations shielded from ocean currents to account for habitat patchiness with increased likelihood to capture genetic disjunction at smaller spatial scales. This is crucial in order to accurately investigate the number of genetic clusters and to avoid misattributing them to interbreeding between farmed escapees and wild individuals. Furthermore, it is imperative to conduct a comprehensive investigation into the mechanisms of local adaptation under high gene flow by employing a population transcriptomics framework. This approach will allow for a conclusive examination of the genetic and expression patterns, as well as an exploration of adaptive divergence at various levels, including sequences, genes, and biological metabolic pathways, among natural populations. This will further establish fundamental data regarding the genetic makeup of broodstock and assess the impact of artificial selection on adaptive polymorphisms, as well as their potential linkage with causal variants underpinning production-related traits in breeding programs.

## Figures and Tables

**Figure 2 genes-14-01870-f002:**
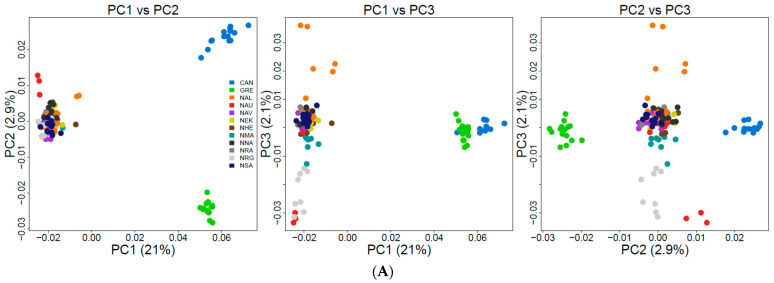
Principal coordinate analysis (PCoA) based on the full (**A**) and 150 genome-wide SNPs (**B**) for the global sampling scheme across the Pan Atlantic.

**Figure 3 genes-14-01870-f003:**
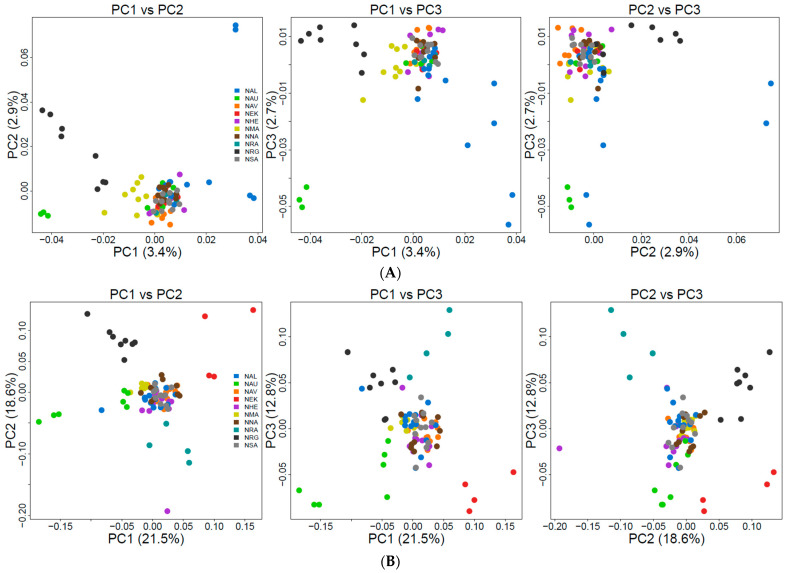
Principal coordinate analysis (PCoA) based on the full neutral (**A**) and 150 genome-wide SNPs (**B**) for the regional sampling scheme in Norway.

**Figure 4 genes-14-01870-f004:**
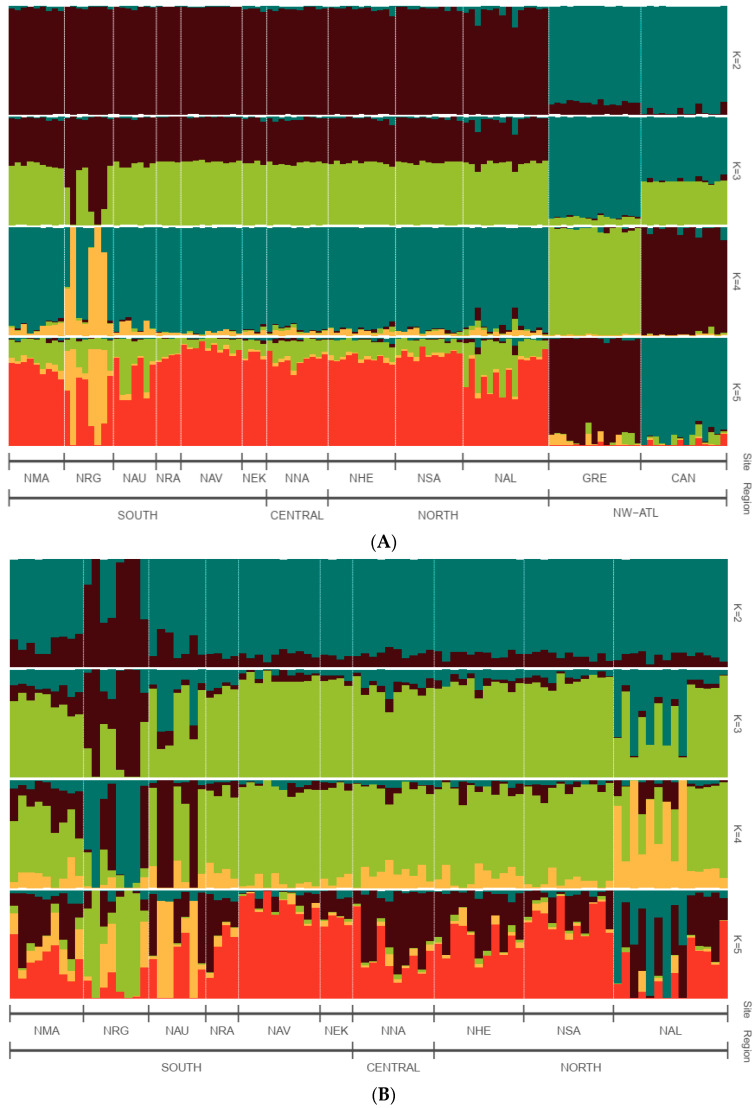
Sparse non-negative matrix factorisation (snmf) clustering analyses for a global (**A**) and regional (**B**) picture of genetic cluster patterns of lumpfish across the north Atlantic for 10 independent runs of K = 2–5 based on the full neutral SNP dataset. NW-ATL, North-western Atlantic Ocean. A single vertical line represents each individual; a black line separates sample sites; the whole sample is divided into *K* colours representing the number of clusters inferred. The colours show the estimated individual proportions of the cluster membership, where each colour represents a genetic cluster.

**Figure 5 genes-14-01870-f005:**
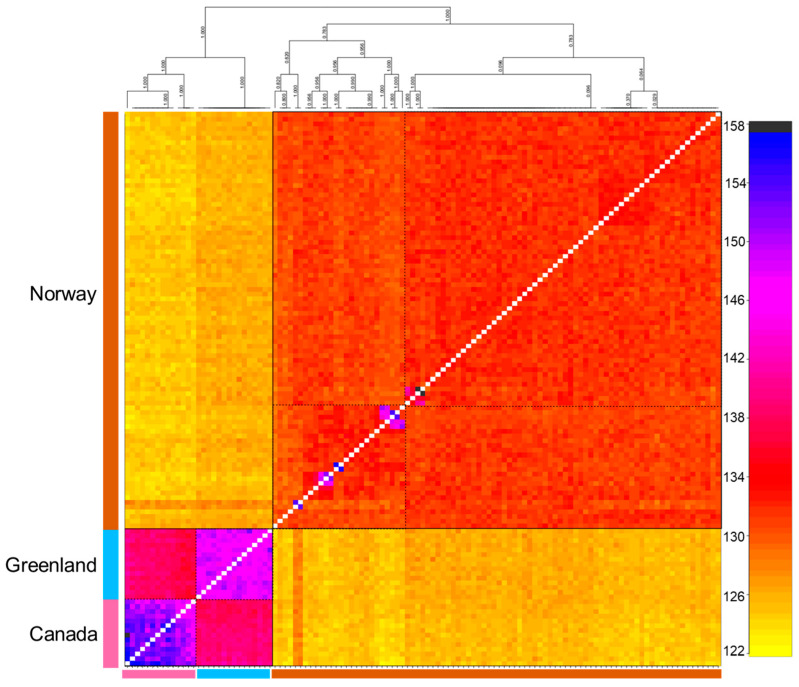
fineRADstructure pairwise co-ancestry matrix and tree based on the global full neutral SNP datasets. The central matric panel shows co-ancestry between genotypes, with light yellow indicating low co-ancestry and darker yellows, oranges, and reds to black indicating progressively higher co-ancestry. Solid black squares indicate the major clades corresponding to ocean basin. Dashed squares indicate the sub-clades.

**Figure 6 genes-14-01870-f006:**
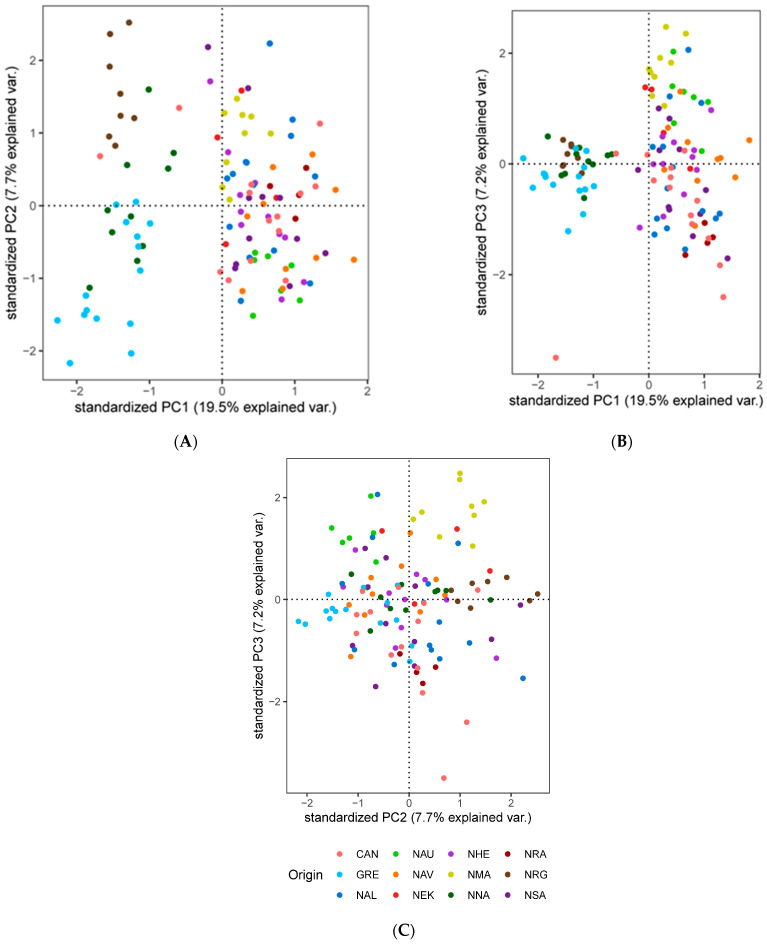
Principal component analysis (PCA) based on the 27 genome-wide CNVs based on PC axes 1 and 2 (**A**), PC axes 1 and 3 (**B**), and PC axes 1 and 3 (**C**) for the global sampling scheme in the North Atlantic.

**Figure 7 genes-14-01870-f007:**
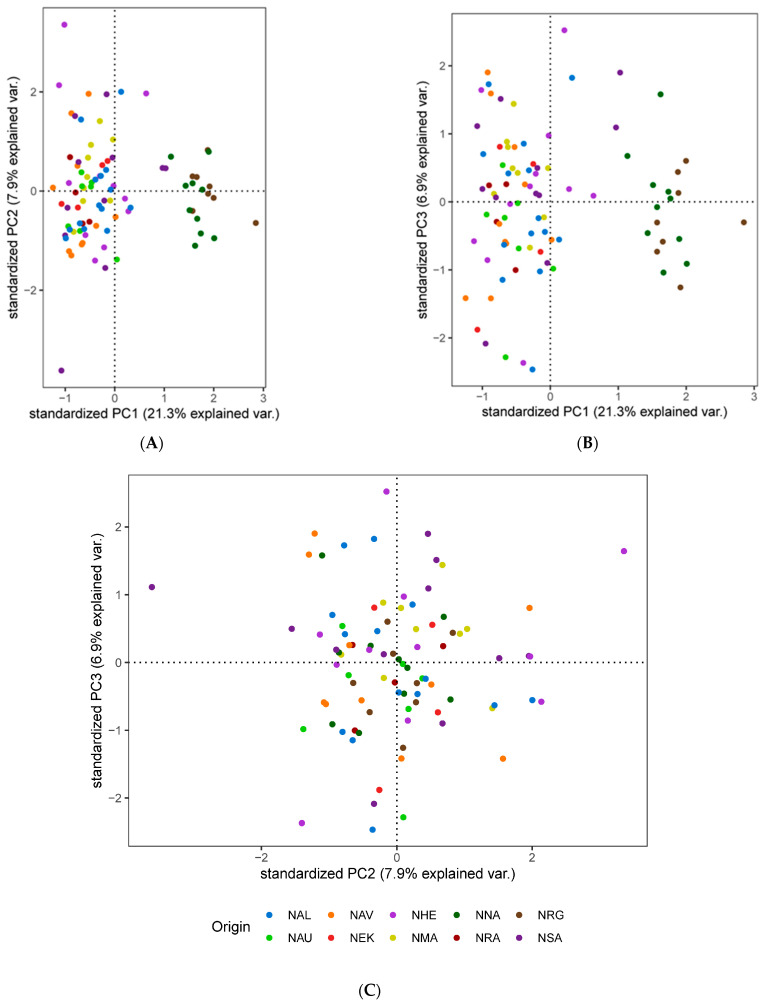
Principal component analysis (PCA) based on the 27 genome-wide CNVs based on PC axes 1 and 2 (**A**), PC axes 1 and 3 (**B**), and PC axes 1 and 3 (**C**) for the global sampling scheme in the North Atlantic for the regional sampling scheme in Norway.

**Figure 8 genes-14-01870-f008:**
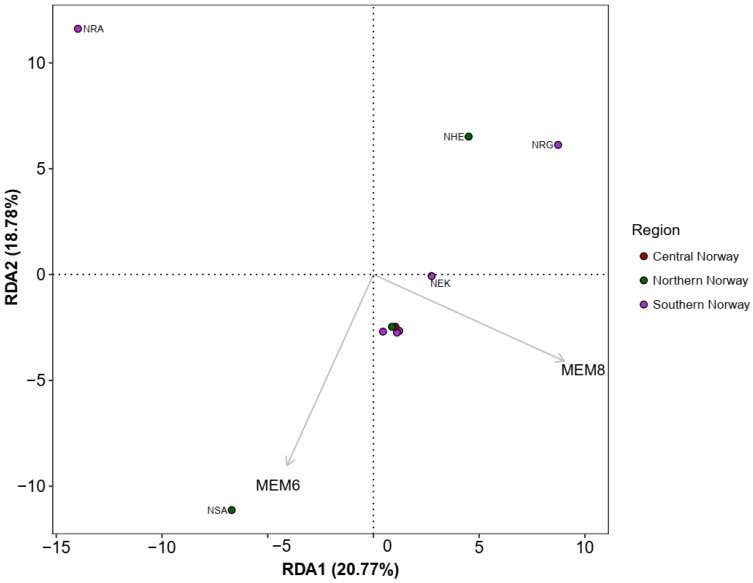
RDA biplot showing the significant Moran Eigenvector Maps (MEMs) predictor variables (arrows) on RDA axes 1 and 2.

**Figure 9 genes-14-01870-f009:**
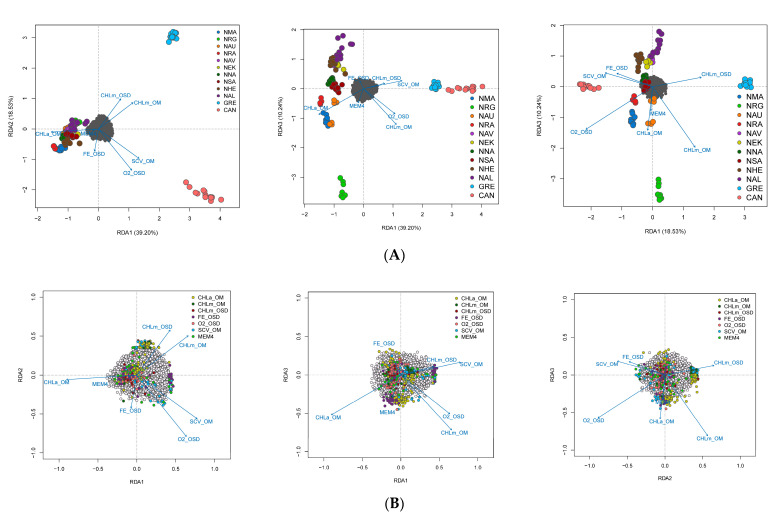
RDA biplot of sampling sites (**A**) and SNPs (**B**) based on RDA axes 1 and 2, RDA axes 1 and 3, and RDA axes 1 and 3 for the global sampling scheme in the North Atlantic.

**Figure 10 genes-14-01870-f010:**
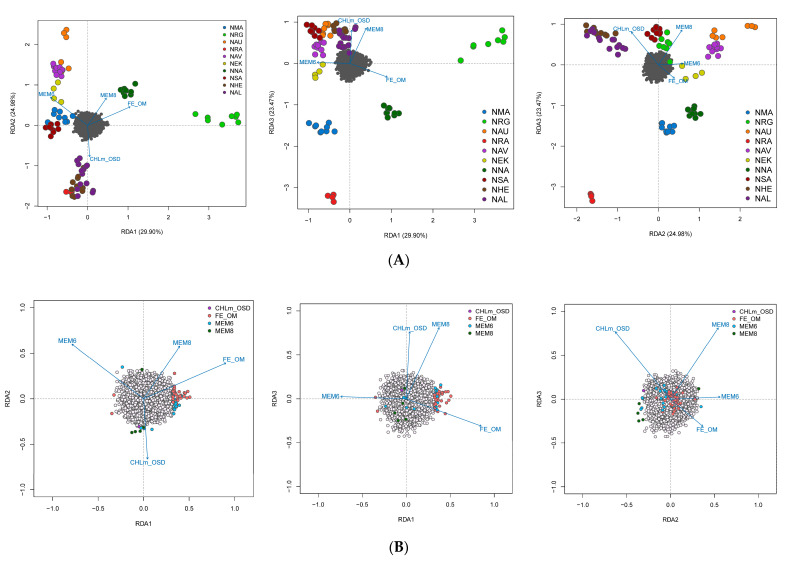
RDA biplot of sampling sites (**A**) and SNPs (**B**) based on RDA axes 1 and 2, RDA axes 1 and 3, and RDA axes 1 and 3 for the regional sampling scheme in Norway.

**Figure 11 genes-14-01870-f011:**
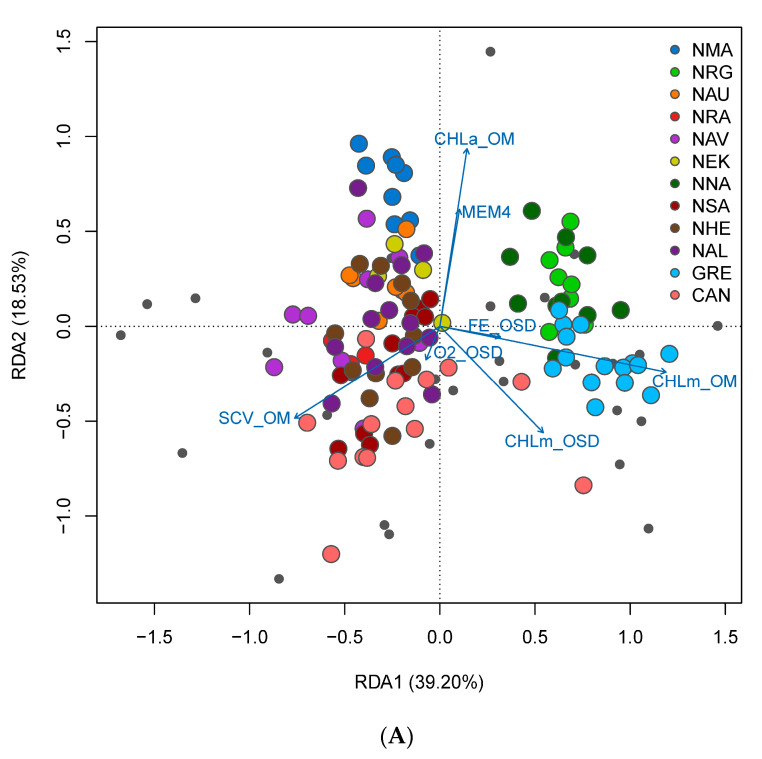
RDA biplot of sampling sites based on RDA axes 1 and 2 for the global (**A**) and regional (**B**) CNV datasets.

**Table 2 genes-14-01870-t002:** Pairwise *F_ST_* (SNPs) and *V_ST_* (CNV) and their confidence interval in brackets among sampling populations of lumpfish estimated using the full neutral SNP and CNV datasets. *F_ST_*-values are below the diagonal and *V_ST_*-values are above the diagonal.

	NMA	NRG	NAU	NRA	NAV	NEK	NNA	NSA	NHE	NAL	GRE	CAN
NMA		0.111 (0.067–0.156)	0.345 (0.232–0.458)	0.134 (0.057–0.212)	0.089 (0.037–0.141)	0.182 (0.11–0.254)	0.159 (0.065–0.252)	0.219 (0.123–0.314)	0.095 (0.056–0.134)	0.046 (0.019–0.072)	0.147 (0.082–0.213)	0.317 (0.213–0.421)
NRG	0.019 (0.017–0.021)		0.127 (0.065–0.188)	0.19 (0.098–0.281)	0.139 (0.074–0.203)	0.243 (0.148–0.338)	0.301 (0.193–0.408)	0.116 (0.065–0.168)	0.078 (0.042–0.115)	0.057 (0.033–0.08)	0.219 (0.13–0.309)	0.163 (0.099–0.227)
NAU	0.009 (0.006–0.012)	0.032 (0.028–0.035)		0 (0–0)	0.2 (0.105–0.295)	0.271 (0.18–0.363)	0.169 (0.062–0.277)	0.272 (0.168–0.377)	0.127 (0.065–0.189)	0.054 (0.02–0.088)	0.103 (0.039–0.166)	0.276 (0.179–0.374)
NRA	0.004 (0.002–0.007)	0.032 (0.028–0.035)	0.01 (0.006–0.015)		0.174 (0.107–0.242)	0.248 (0.148–0.349)	0.325 (0.212–0.437)	0.181 (0.087–0.275)	0.238 (0.151–0.325)	0.077 (0.035–0.119)	0.121 (0.048–0.195)	0.287 (0.189–0.385)
NAV	0.007 (0.005–0.009)	0.029 (0.026–0.031)	0.015 (0.012–0.018)	0.006 (0.003–0.009)		0.185 (0.084–0.286)	0.198 (0.125–0.271)	0.154 (0.068–0.24)	0.207 (0.132–0.283)	0.115 (0.058–0.172)	0.138 (0.061–0.216)	0.226 (0.137–0.315)
NEK	0.003 (0–0.005)	0.03 (0.027–0.034)	0.008 (0.004–0.013)	0.005 (0.001–0.009)	0.001 (−0.002–0.005)		0.191 (0.083–0.3)	0.258 (0.144–0.372)	0.123 (0.07–0.176)	0.1 (0.039–0.161)	0.126 (0.054–0.198)	0.313 (0.198–0.428)
NNA	0.004 (0.003–0.006)	0.027 (0.025–0.03)	0.011 (0.009–0.014)	0.002 (−0.001–0.005)	0.006 (0.004–0.008)	−0.002 (−0.004–0.001)		0.076 (0.037–0.114)	0.087 (0.051–0.123)	0.088 (0.05–0.126)	0.087 (0.046–0.128)	0.308 (0.208–0.409)
NSA	0.006 (0.004–0.008)	0.03 (0.027–0.033)	0.016 (0.013–0.019)	0.008 (0.005–0.011)	0.004 (0.002–0.005)	0.005 (0.002–0.008)	0.005 (0.003–0.006)		0.27 (0.162–0.378)	0.094 (0.049–0.138)	0.057 (0.013–0.101)	0.331 (0.22–0.442)
NHE	0.005 (0.004–0.007)	0.026 (0.024–0.029)	0.014 (0.011–0.017)	0.003 (0.001–0.006)	0.005 (0.003–0.007)	0.001 (−0.002–0.003)	0.002 (0–0.003)	0.003 (0.001–0.004)		0.098 (0.064–0.131)	0.113 (0.062–0.163)	0.158 (0.094–0.222)
NAL	0.008 (0.007–0.01)	0.031 (0.028–0.033)	0.015 (0.012–0.018)	0.003 (0.001–0.006)	0.01 (0.008–0.012)	0.002 (−0.001–0.005)	0.007 (0.005–0.008)	0.008 (0.007–0.01)	0.006 (0.005–0.008)		0.061 (0.036–0.086)	0.179 (0.12–0.238)
GRE	0.118 (0.114–0.122)	0.141 (0.137–0.146)	0.134 (0.129–0.138)	0.132 (0.126–0.137)	0.127 (0.122–0.131)	0.124 (0.118–0.129)	0.122 (0.118–0.126)	0.126 (0.122–0.131)	0.123 (0.119–0.127)	0.118 (0.114–0.122)		0.048 (0.023–0.074)
CAN	0.153 (0.148–0.159)	0.182 (0.176–0.187)	0.17 (0.164–0.176)	0.176 (0.169–0.182)	0.165 (0.159–0.171)	0.165 (0.158–0.171)	0.156 (0.151–0.162)	0.163 (0.158–0.168)	0.157 (0.152–0.163)	0.152 (0.147–0.157)	0.068 (0.065–0.071)	

## Data Availability

The data presented in this study are available on request from the corresponding author. The data are currently not publicly available but are pending release on curated data repository. The data will be released upon acceptance of this manuscript.

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
