# Peer review of "Genomic Signatures of Local Adaptation under High Gene Flow in Lumpfish—Implications for Broodstock Provenance Sourcing and Larval Production"

_genes, 2023, doi:10.3390/genes14101870_

Round 1

Reviewer 1 Report

The reviewer feels that the introduction in the preface is too complicated and recommends that the content be streamlined appropriately.

Perhaps this part of "2.1. Study Species" should be described in the introduction? The reviewer was confused by this.

The first person "we" is used in many places in the manuscript, and the reviewers suggest that you try not to use similar personal pronouns in the manuscript.

If the map in Figure 1 was not drawn by the author, the source should be indicated.

There should be spaces between numbers and units, which need to be checked and revised throughout the text.

Make sure that all works cited in the text are used correctly, that the presentation is consistent and that correct information is given in the reference list.

Moderate editing of English language required

Reviewer 2 Report

Dear Editors,

The manuscript entitled " Genomic Signatures of Local Adaptation under High Gene Flow in Lumpfish—Implications for Broodstock Provenance Sourcing and Larval Production" describes a comprehensive analysis of genomic data from lumpfish populations in the North Atlantic, with a particular focus on understanding genetic diversity, local adaptation, and the relationship between genetic variation and environmental factors in the lumpfish populations inhabiting Norwegian coast. The authors utilized a high-throughput sequencing approach to generate genomic data from lumpfish populations. They applied various statistical methods to identify SNPs under balancing and divergent selection, eventually creating a dataset of adaptive SNPs and to make functional annotation of environment-associated SNPs. Moreover, the applied rich analytical methods aimed to: (1) investigate genetic diversity and differentiation, (2) population genetic structure and connectivity, (3) spatial structure and environmental associations between lumpfish populations in the in the North Atlantic.

The obtained results of the reviewed study have several important implications:

(1) The identification of genetic clusters and local adaptation patterns can help inform conservation efforts and management strategies for lumpfish populations. It may also aid in addressing genetic impacts caused by escapees from aquaculture facilities.

(2) Understanding the genetic basis of adaptation to environmental factors can provide valuable insights into the mechanisms that enable lumpfish to thrive in different habitats. This knowledge may inform future breeding and selection programs in aquaculture.

(3) The dataset of SNPs and CNVs generated in this study can serve as a valuable genetic resource for future research on lumpfish and related species.

(4) The findings may have practical applications in the aquaculture industry, especially in Norway, where lumpfish are used as cleaner fish. Genetic-informed broodstock selection and management could enhance the effectiveness of lumpfish in controlling sea-lice infestations in salmon farms.

The introduction chapter of the manuscript provides an overview of the aquaculture industry, specifically focusing on the Atlantic salmon mariculture and the use of cleaner fish, such as lumpfish, to control sea lice infestations. The authors also introduce the central themes of their research, including genetic diversity, local adaptation, and the implications for lumpfish aquaculture.

The Materials and Methods chapter provides a comprehensive and detailed description of the research methods and procedures used in the study. Applied methods are impressive and no issues have been detected. Overall, the chapter appears to be well-structured and rigorous. It provides a clear understanding of how the study was conducted, and the use of multiple methods and robust data filtering procedures enhances the credibility of the research findings.

The Results chapter provides a robust, comprehensive and detailed data, covering various aspects of genetic diversity, differentiation, population structure, and environmental associations.

The Discussion chapter integrates the current findings with previous research effectively, resolving conflicting evidence and providing a comprehensive view of the state of knowledge regarding lumpfish population genetics. It offers valuable insights into the complex interplay of local adaptation and gene flow in marine species, specifically lumpfish, and provides a strong conclusion that ties together the key takeaways from the research. While the chapter provides a strong analysis and interpretation of the current study's results, it could benefit from a brief discussion of potential future research directions, especially in terms of further exploring the mechanisms of local adaptation and its application in aquaculture and conservation.

Overall, the reviewed study have relevance for both conservation efforts and the sustainable aquaculture of lumpfish. The substantive content of the manuscript is correct, and no major issues were found. Some issue I have found with sample numbers but the specific details are described in the attached pdf file. The language used is technical but clear, suitable for the intended scientific audience.

Best regards,

Reviewer 3 Report

Review comments on Maduna et al

Genomic Signatures of Local Adaptation under High Gene Flow in Lumpfish—Implications for Broodstock Provenance Sourcing and Larval Production

The aim of the study was to investigate spatial patterns of genomic variation in lumpfish sampled at various locations along the Norwegian coast, Greenland and Canada. The paper describes in extensive detail how the genetic data was quantified from the fin clip or muscle samples from each sampling location. The authors also described in detail the variety of data management and statistical techniques used for analysis of the genetic data. The results of the study showed that there was broad genetic variation between the western and eastern Atlantic populations, as well as spatial variation in genetic structuring between the southern and northern Norwegian populations.

Overall, the data presented in the paper achieves the study aims and provides an original contribution to knowledge about the population genetics of lumpfish in Norwegian waters and will provide a sound basis for management broodstock used for captive breeding of lumpfish used for parasite control at salmon farms. The extensive statistical analysis of the genetic data was appropriate to address the aims of the study. As such, I consider that a revised paper addressing the comments detailed below would be suitable for publication.

Specific comments on the manuscript are detailed below.

Page 2 par 1

Aquaculture production alleviate commercial fishing pressure… 

Change alleviate to alleviates.

Tasmania is a state of Australia and is not a country.

Page 2 par 2

development of chemical resistant strain of sea lice [8,18,19].

Change spelling of strain to strains.

Page 7 section 2.3.2

We removed reads with an uncalled bases (-c) and discarded reads with low quality scores (-q) with a default sliding window of 15% of the length of the read and raw Phred score of 20.

The authors should revise this sentence to clarify the text.

Page 9 section 2.3.4

…we defined an false discovery rate (FDR) q-value threshold of 0.05…

Change spelling of an to a.

Page 11 par 1

for species with a plantonic larval phase [116– 119].

Change plantonic to planktonic.

Page 13 section 2.7

First, we annotated environment-associated SNP based on the NCBI annotation of the reference genome of the lump[1]fish using SNPEFF v.5.1 [142], and we retrieved all the predicted genes falling within the ±50 kilobases (kb) window.

The authors should revise this sentence to clarify the text.

Page 13 section 3.1

Processing of raw Illumina data by the program process_radtags we recovered a high percentage of retained reads..

Delete we after radtags.

Page 22 par 1

However, within Norway we detect no significant effect of IBD with the regional dataset…

Change spelling of detect to detected.

Page 26 section 4.1

The putative genomic regions under divergent selection that we uncovered revealed that sampling populations from Norway…

Change sampling to sampled.

Page 28 section 4.5

Our reginal sampling regime allowed…

Change spelling of reginal to regional.

Reference No 16

16. Johnson, S.C.; Constible, J.M.; Richard, J. Laboratory investi-gations on the efficacy of hydrogen peroxide against the sal-mon louse Lepeophtheirus salmonis and its toxicological and histopathological effects on Atlantic salmon Salmo salarand Chinook salmonOncorhynchus tshawytscha. Dis. Aquat. Org. 1993, 17, 197–204.

Correct the formatting of this reference

121. Barbosa, A.M. fuzzySim: applying fuzzy logic to binary similarity indices in ecology. Methods in Ecology and Evolution. 2015, 6, 853–858.

Abbreviate the journal title for this reference similar to reference No 108.

No issues with the quality of English
